# Mapping and Statistical Analysis of NO$_2$ Concentration for Local Government Air Quality Regulation

**Jieun Ryu [1], Chan Park [2] and Seong Woo Jeon [1,\*]**

1   Department of Environmental Science and Ecological Engineering, Korea University, Seoul 02841, Korea
2   Department of Landscape Architecture, University of Seoul, Seoul 02504, Korea
\*   Correspondence: eepps_korea@korea.ac.kr; Tel.: +82-2-3290-3043

**Abstract:** With the growing interest in healthy living worldwide, there has been an increasing demand for more accurate measurements of the concentrations of air pollutants such as NO$_2$. In particular, analyzing the characteristics and sources of air pollutants by region could improve the effectiveness of environmental policies applied in accordance with the environmental characteristics of individual regions. In this study, a detailed nationwide NO$_2$ concentration map was generated using the cokriging interpolation technique, which integrates ground observations and satellite image data. The root-mean-square standardized (RMSS) error for this technique was close to 1, which indicates high accuracy. Using spatially interpolated NO$_2$ concentration data, an administrative unit map was generated. When comparing the data for four NO$_2$ data sources (observation data, satellite image data, detailed national data interpolated using cokriging, and NO$_2$ concentrations averaged by an administrative unit based on the interpolated NO$_2$ concentration data), the average concentrations were highest for remote sensing data. Land use regression (LUR) models of urban and non-urban regions were then developed to analyze the characteristics of the NO$_2$ concentration by region using NO$_2$ concentrations for the administrative units.

**Keywords:** urban forest; nitrogen dioxide; interpolation; cokriging; NO$_2$ concentration map; satellite image; land use regression model; county level

## 1. Introduction

The air quality is deteriorating globally at an alarming rate due to increasing industrialization and urbanization. In particular, the concentration of nitrogen dioxide (NO$_2$) is increasing significantly due to anthropogenic activities [1,2], with most NO$_2$ being generated by road vehicles and industrial activities [3–7]. As the global population becomes more health conscious, various studies have been conducted to determine the effect of NO$_2$ concentration on human health [8]. High NO$_2$ concentrations in urban areas cause bronchial and lung cancer [9] and have severe effects on asthmatic patients [10]. In 15 EU countries, the average social medical costs due to NO$_2$ pollution were estimated to be approximately US $4.85 billion per ton [11].

To reduce these social costs, many countries regulate NO$_2$ concentration levels using environmental policies that target NO$_2$ reduction. For example, the EU has established an integrated environmental policy agreement for the transport, industrial, and energy sectors to improve air pollution at national, regional, and local levels [12]. In addition, since 2013, China has sought to install selective catalytic reduction (SCR) equipment in power plants to establish emissions standards to reduce NO$_2$ levels through the Air Pollution Prevention and Control Action Plan [13].

To establish effective environmental regulations that reduce the impact of NO$_2$, accurate data on NO$_2$ concentrations within administrative units on a county scale is essential. Because environmental

regulations are applied to these units, it is important to be able to accurately predict the effect of these regulations and monitor their effect. In addition, population and $NO_2$ emission source data is collected by administrative units, and this data is necessary for decision-making related to environmental policy. In China, the same guidelines regarding $NO_2$ regulations are applied nationwide [14] and several studies have analyzed changes in $NO_2$ concentration and regulatory effects in administrative units [14,15].

However, there is lack of accurate $NO_2$ concentration data for administrative units. Three types of data have been used to produce $NO_2$ concentration maps for administrative units: spatial data predicted using actual observed point-type data, spatial data from satellite images, and statistical models. Observational data is highly accurate, but information is only available for the area surrounding an observatory [16]. To overcome this restriction, some researchers have used satellite data of lower spatial resolution rather than observational data, as it offers the use of a wider range of cell-based data [17]. However, the accuracy of remote sensing data is weakened by the assumptions associated with satellite calculation algorithms, cloud and surface reflections, and the absence of vertical distribution data [16]. Another method for predicting the distribution of $NO_2$ concentrations uses air quality modeling and statistical interpolation based on observation point data [18–20]. Air quality models are also used to overcome the limitations of observational and satellite data [16,21]. However, atmospheric modeling is disadvantageous in that although the spatial resolution is high, it is difficult to judge the level of uncertainty [22] and it is expensive. As a result, an increasing number of studies have investigated the spatial interpolation of observational and satellite data using cost-effective geostatistical methods, with highly accurate results [23].

The aim of this study is to generate $NO_2$ concentration data for administrative units and analyze $NO_2$ characteristics based on this data as a means to effectively monitor the effects of environmental regulations. To achieve this, detailed $NO_2$ concentration data was generated for each map cell along with observational data and satellite images. Based on this data, an accurate $NO_2$ concentration map for administrative units was constructed, and seasonal concentrations were predicted using land use regression analysis. The LUR model is a statistical method that is widely used for analyzing the characteristics of air pollutant concentrations, because it has the advantage of being able to grasp the environmental influence on the $NO_2$ concentration spatially [24–26].

## 2. Method

### 2.1. Study Area

Asia is developing rapidly and as a consequence, high levels of air pollutants are emitted in this region [27,28]. In Asia, especially in China, Korea and Japan, $NO_2$ concentrations are higher than in other regions. The Republic of Korea (hereafter, South Korea), located in East Asia at 33°–43°N and 124°–132°E (Figure 1) has a population of ~51,696,216 people (based on data from the Ministry of Government Administration and Home Affairs in 2017) and has a land area of ~99,720 km². Due to the North Pacific high-pressure system in the summer, southeast and/or southwest winds are strong. Based on this atmospheric circulation, the amount of air pollutants from neighboring countries is very high [1].

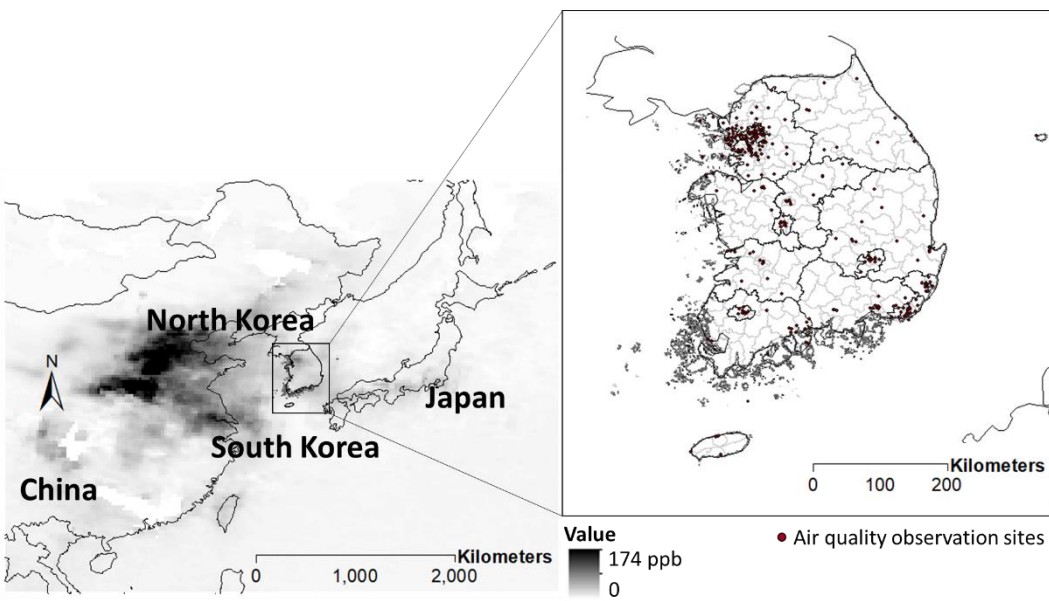

**Figure 1.** Map of East Asia showing the distribution of $NO_2$ concentrations. Inset: Locations of the $NO_2$ observation points in South Korea.

In South Korea, policies aim to precisely measure the concentration of air pollutants and reduce the volume of air pollutants, including $NO_2$. Previous government policies have focused on the accurate measurement of $NO_2$ levels; however, the government has recently made efforts to reduce $NO_2$ levels by actively legislating mitigation measures. Based on the Clean Air Conservation Act (Law No. 14532, revised on November 17, 2017), the government has selected carbon monoxide (CO), nitrogen oxides ($NO_x$), sulfur oxides ($SO_x$), dust (e.g., $PM_{2.5}$ and $PM_{10}$), volatile organic compounds (VOCs), and $NH_3$ (ammonia) as target air pollutants, and the Korea Environment Corporation (KEC) operates and manages a national air pollution monitoring network that monitors these substances. In terms of air quality management, the KEC manages air quality based on the overall air quality and the remote monitoring of the type and amount of pollutants discharged into the chimney. In addition, the Seoul Metropolitan Government and the national government have introduced a low-pollution project for old vehicles, while the Ministry of Environment revised the laws and regulations related to the Clean Air Conservation Act No. 35 (the collection of emission levies) in June 2018 to impose levies on nitrogen oxide ($NO_x$) emission from industrial sites. There are many policies related to fine dust in Korea, but "Comprehensive Measures for Fine Dust Management" jointly announced by various ministries on 26 September 2017 is representative. However, to link observed data with these policies, it is necessary to analyze $NO_2$ levels based on detailed national $NO_2$ concentration data and the spatial characteristics of observational data.

## 2.2. Generation of $NO_2$ Concentration Data

### 2.2.1. Monitoring $NO_2$ Levels

In South Korea, the KEC has installed observation stations at specific locations to measure $NO_2$ levels. We used 2010 data for this study, at which time 240 stations had been installed, fewer than the 264 stations available in 2016 [29]. By 2010, KEC had installed 19 stations in the suburbs (compared with 19 stations in 2016), 2 stations as part of the national background concentration network (3 stations in 2016), and 33 stations as part of the roadside monitoring network (38 stations in 2016). In this study, monthly average $NO_2$ concentration data from December 2009 to November 2010 was downloaded from the Air Korea website (http://www.airkorea.or.kr/index).

2.2.2. Satellite Images Used to Monitor $NO_2$

The vertical column (VC) density of $NO_2$ is measured using three different satellite sensors: the Global Ozone Monitoring Experiment-2 (GOME-2), the Ozone Monitoring Instrument (OMI), and the Scanning Imaging Spectrometer for Atmospheric Chartography (SCIAMACHY) [30]. We used GOME-2 images from the Meteorological Operational–A (MetOp-A) satellite because they provide data for various time periods. The ground pixel size of this satellite data is 80 × 40 km, and the global revisit time is 1.5 days. GOME-2 produces both monthly and daily images. We used the monthly average $NO_2$ concentration data from GOME-2 for the period from December 2009 to November 2010, which were provided by the National Aeronautics and Space Administration (NASA) (See Figure A1).

Monthly data from December, January, and February were used for the winter season and data from June, July, and August were used for the summer season. Because $NO_2$ emissions vary between seasons, seasonal analysis is needed [31]. In order to minimize the influence of weather, correlation analysis between observational data and satellite images was performed first, and satellite images with high correlations were used in this research. Pearson correlation analysis was performed between the observed monthly data and satellite imagery data from 2009 and 2013. Based on the results, we decided to use February 2010 to represent winter data (a correlation coefficient of 0.674, $p = 0.001$) and August 2010 to represent summer data (a correlation coefficient of 0.673, $p = 0.001$).

2.2.3. Geostatistical A Spatial Analysis: Cokriging

To generate detailed nationwide $NO_2$ concentration data, we used the cokriging method, which is a form of geostatistical spatial analysis. It can improve the accuracy of estimations by employing spatial interpolation based on the relationship between environmental data and a particular reference value [32,33]. The final product derived from the cokriging method is a raster, which has pixel values. In this study, spatial interpolation was conducted based on observational data using satellite images as supplementary data. Daily and monthly $NO_2$ levels in the suburban atmosphere, country background, roadside atmosphere, and urban atmosphere provided by Air Korea were used as the primary data for interpolation.

Before implementing the cokriging method, a semivariogram was constructed to see whether the data was skewed and to confirm spatial autocorrelation. A semivariogram is a measure of the similarity between data points at a certain distance based on spatial correlation [34]. When the number of data points spaced apart by separation distance $h$ is $n$, a semivariogram can be calculated using Equation (1), where $r$ is the value of the semivariogram, $z$ is the value of the data at arbitrary point $x$, and $h$ is the distance between the data points [35]:

$$\text{r(h)} = \frac{1}{2n} \sum_{i=1}^{n} [z(x_i) - z(x_i + h)]^2 \tag{1}$$

Since only satellite image data is used as a sub-parameter based on the observation data, it can be represented as a simple cokriging equation like Equation (2) by using the main variable $Z_i$ and a secondary variable $u_i$. $m$ is the number of data points for the secondary variable, and $k_j$ and $\lambda$ are the weights of the data used [24,32]:

$$\text{Z} = \sum_{i=1}^{n} \lambda_i Z_i + \sum_{j=1}^{m} k_j u_j \tag{2}$$

Cross-validation can be used to judge prediction performance [31]. The accuracy of the estimates was verified using the root-mean-square standardized (RMSS) error between the observed values and distributions generated through cokriging. In Equation (3), $P$ and $O$ represent the estimated and observed values at point $i$, respectively, and $Q$ is the mean. The total number of samples is

represented by *n*. The closer the RMSS value is to 1, the more accurate the difference between the original observation and interpolated value.

$$\text{RMSS} = \sqrt{\frac{1}{n}\sum_{i=1}^{n}\{(P_i - O_i)/Q_i\}^2} \tag{3}$$

### 2.3. Comparison of Data Characteristics by Spatial Unit

NO$_2$ levels may differ from season to season due to differences in demand for automobiles and heating, which directly affect the concentration of NO$_2$. According to this, the accuracy of the concentration data and the differences between sources was analyzed for data from both summer and winter. To compare the spatial seasonal differences in NO$_2$ concentration, the mean, maximum, minimum, median, variance, and standard deviation of the NO$_2$ concentration were calculated by region and season using the zonal statistic function of ArcGIS 10.2. The datasets from satellite images, observations, spatially interpolated NO$_2$ concentrations, and the reprocessing of spatially interpolated NO$_2$ concentrations by administrative unit were compared and statistically analyzed by comparing the data range and standard deviation.

### 2.4. Prediction of NO$_2$ Concentrations by Administrative Unit Using A Land Use Regression Model

To analyze the sources of NO$_2$ in each region, land use regression (LUR) models were developed using the relationship between NO$_2$ concentration and environmental variables. The LUR models were developed for urban and non-urban areas in order to identify the factors with major influence on the increase in NO$_2$ concentrations in the administrative units. We classified urban and non-urban areas using land cover categories as administrative units. The NO$_2$ levels in urban regions are more variable than in rural areas due to micro-meteorological factors [36] and a high proportion of artificial surfaces in urban settings. Based on the land cover data, residential, industrial, commercial, recreational, traffic, and public areas were all classified as artificial surfaces. The non-urban area was divided into agricultural areas (rice fields, fields, facility plantations, orchards, and other plantations), artificial grasslands, natural grasslands, forest areas (broadleaved forest, coniferous forest, and mixed forest), wetlands (inland wetland and coastal wetland), natural springs, and inland waters were classified as non-urban areas.

South Korea is divided into 252 administrative districts (referred to as *gu* in Korean). Urban areas contain a total of 51 of these districts, and it has the highest ratio of impervious areas compared to other regions (Figure 2). Most of the metropolitan administrative districts are in Seoul and Busan. Some parts of Gyeonggi, Daegu, Incheon, Gwangju, Ulsan, Mokpo, and Gwangyang that are highly industrialized are considered metropolitan cities. These cities are classified as high-density, developed, urban areas. Non-urban regions contain a total of 201 administrative districts. Although there is human activity in these non-urban districts, it is much less intensive than that in urban areas.

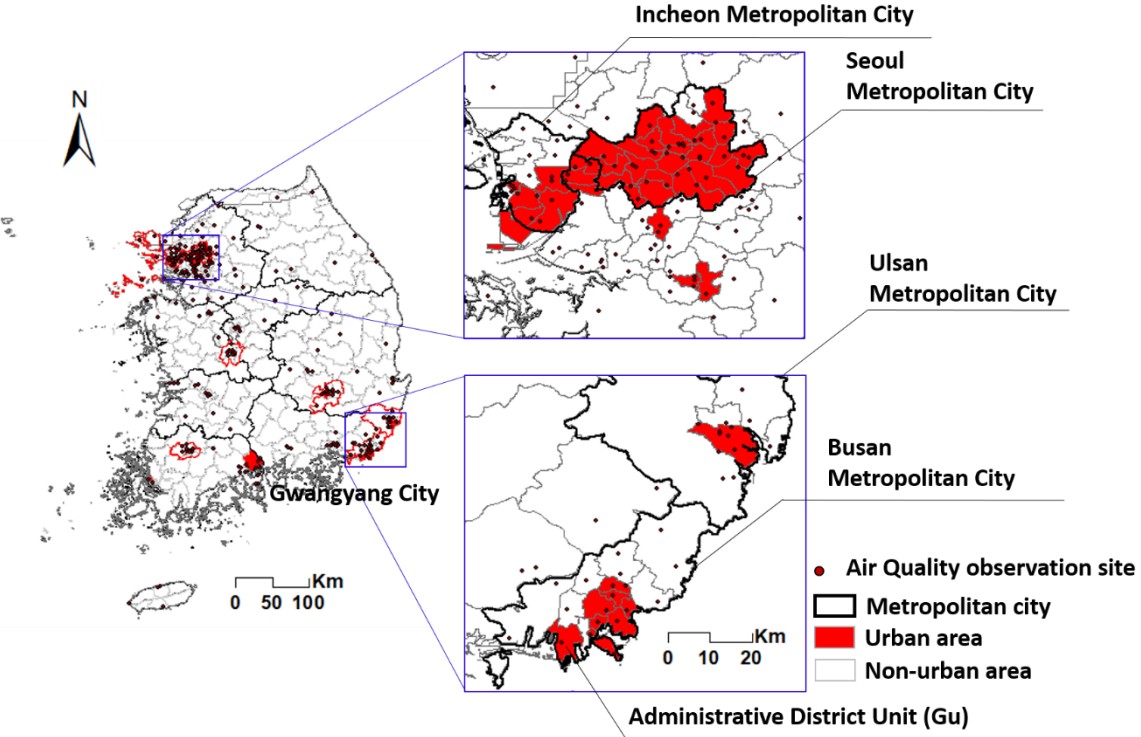

**Figure 2.** Administrative units in urban areas within South Korea.

LUR modeling predicts air pollutant concentrations based on surrounding land use and environmental properties [24–26,37,38]. To develop this model, we used spatialized environmental databases as predictor variables. In this study, linear regression equations were developed using the enter procedure, as shown in Equation (4). The environmental factors affecting the concentrations of $NO_2$ are indicated by relation between y and x. $\alpha_n$ are the correlation coefficients ($n = 1 \ldots n$) between the environmental factors and $NO_2$ concentrations, $Xn$ is an environmental factor ($n = 1 \ldots n$), and *c* is a constant. The linear regression model is able to explain the effect of the environmental variables on $NO_2$ concentrations because the correlation between the two is clear. The LUR models were developed according to the regional characteristics of the $NO_2$ concentration.

$$y = a_1 \times x_1 + a_2 \times x_2 + a_3 \times x_3 + \ldots + a_n \times x_n + c \tag{4}$$

Previous studies have used spatial data on traffic, population, land use, and physical geography as predictor variables [4,5,35–43]. A total of 18 environmental variables have been used as predictor variables (Table 1). Because the normalized difference vegetation index (NDVI) and $NO_2$ concentrations exhibit significant seasonal variation, they were divided by season using a dummy variable that bears the value of 1 for summer and 0 for winter. This is because the purpose of this study is to quantitatively analyze the differences in the seasonal and regional characteristics of $NO_2$ concentrations. The emission of air pollutants varies depending on land use [39], and the concentration and mechanisms of $NO_2$ emission may vary depending on the source [4,5,39]. In previous studies, $NO_2$ has been considered a traffic-originating pollutant [40–42], so the length of the roads within a district was used as a variable (Road_Length) in this study.

In relation to human activity, including power consumption and industrial activities [5], we used the proportions of residential area (R_Res), industrial area (R_Ind), commercial area (R_Com), cultural and sports recreation area (R_Cul), public facility area (R_Pub), and urban area (R_Urban) as prediction variables. Population is an indicator of urban growth [38], and air quality is closely related to urbanization and population growth, so population was used as an indicator in this study (R_Pop).

Because the population and road length have a significant effect on $NO_2$ concentrations, road length per unit of population was also used as a variable (RL_Pop).

In South Korea, forests accounted for about 63.7% of the total national land area [43] in 2010. Because the topography of the country is complex, and the influence of monsoons on the temporal and spatial variability of most meteorological factors is significant, it is necessary to consider topographical variables when quantitatively analyzing climate characteristics [44,45]. Therefore, a digital elevation model (DEM) was used as an environmental variable (DEM_std) [46]. Forest and green spaces do not generate $NO_2$ because they contain almost no emission sources, and they promote less turbulent air flow in the surrounding areas. These areas are thus necessary to prevent deterioration to air quality [47]. In this study, the proportion of coniferous forest (R_Coni), deciduous forest (R_Deci), mixed forest (R_Mixed), and whole forest (R_Forest) as dominant species and the NDVI (NDVI) as a measure of live vegetation cover were used as predictor variables. Because the NDVI is pixel-based, averages were used for each administrative area. The NDVI is useful as a representative indicator of green areas because only vegetated areas have values of 0 or more.

**Table 1.** Description of potential predictor variables.

| | Predictor Variable | Name | Effect | Other Comments |
|---|---|---|---|---|
| **Seasonal factors** | | | | |
| | Season | *Season* | | Summer = 1, winter = 0 |
| **Emission factors** | | | | |
| | Percentage of residential area | *R_Res* | + | Residential area by administrative district/Area of each administrative district *100 |
| | Percentage of industrial complex area | *R_Ind* | + | Industrial complex area by administrative district/Area of each administrative district *100 |
| | Percentage of commercial area | *R_Com* | + | Commercial area by administrative district/Area of each administrative district *100 |
| | Percentage of cultural recreation area | *R_Cul* | + | Area of cultural recreation area by administrative district/Area of each administrative district *100 |
| | Percentage of public facilities area | *R_Pub* | + | Area of public facilities by administrative district /Area of each administrative district *100 |
| | Percentage of urbanization area | *R_Urban* | + | Area of urbanization by administrative district /Area of each administrative district *100 |
| | Length of roads | *Road_Length* | + | Length of roads (km)/Area of each administrative district (km$^2$) |
| | Percentage of population | *R_Pop* | + | Population/Area of each administrative district *100 |
| | Road length per unit population | *RL/Pop* | + | Road_Length/R_Pop |
| **Reduction factors** | | | | |
| | Percentage of paddy field | *R_Paddy* | - | Area of paddy fields within an administrative district /Area of each administrative district *100 |
| | Percentage of field | *R_Field* | - | Area of fields within administrative district /Area of each administrative district *100 |

**Table 1.** *Cont.*

| | Predictor Variable | Name | Effect | Other Comments |
|---|---|---|---|---|
| | NDVI | *NDVI* | - | Using zonal statistics of ArcGIS, we derive the monthly average NDVI data of the Moderate Resolution Imaging Spectroradiometer (MODIS) in the cell using the average for the administrative district |
| | Coniferous forest ratio | *R_Coni* | - | Coniferous forest area by administrative district / Area of administrative district *100 |
| | Deciduous forest ratio | *R_Deci* | -/+ | Deciduous forest area by administrative district /Area of administrative district *100 |
| | Mixed forest ratio | *R_Mixed* | -/+ | Mixed forest area by administrative district /Area of administrative district *100 |
| | Forest ratio | *R_Forest* | - | Forest area by administrative district /Area of administrative district *100 |
| **Topographical factors** | | | | |
| | DEM | *DEM_std* | - | Using the zonal statistics of ArcGIS, we derive the standard deviation value of the DEM for each cell as the average value by administrative unit |

The zonal statistics of ArcGIS 10.1.3 were used to calculate the averages within the districts. After that, Pearson correlation analysis was performed using SPSS 16.0 to determine the correlations between $NO_2$ concentrations and the 18 predictor variables and thereby ascertain the final proxy variables to be used in the LUR models. The variables were selected on the following bases: 1) There are correlations between $NO_2$ concentration and the environmental variables at a $p < 0.01$ significance level, as per a Pearson correlation analysis; Pearson's correlation coefficients were calculated to demonstrate the linear relationship between two variables with a significance level of <0.05 [48]. 2) When the correlation between the variables is high (correlation coefficients over 0.5), a parameter having a high correlation with the $NO_2$ concentration was selected. Pearson correlation coefficients are used widely to examine the correlation between criteria [49]. R ≥ ±0.5, ±0.25 ≤ R < ±0.5, and 0 < R ≤ ±0.25 indicate strong, moderate, and weak positive correlations respectively [50].

## 3. Results

### 3.1. Nationwide $NO_2$ Concentration

The $NO_2$ levels based on spatial interpolation for February (winter) and for August (summer) are presented in Figure 3A,B, respectively. The $NO_2$ concentration data generated using the spatial interpolation method is also reproduced for individual administrative units in Figure 4. Comparing Figures 3 and 4, it can be seen that the trends in $NO_2$ levels were consistent irrespective of whether they were measured by grid or administrative units.

When the spatial distribution of $NO_2$ was compared by season, it could be seen that high $NO_2$ concentrations were more common in winter than in summer. $NO_2$ concentration exceeding 40 parts per billion (ppb; the limit according to WHO's 2005 standards) was found over about 26.14% of the total land area in February, but only over about 0.01% of the land area in August. In February, the concentration of $NO_2$ was higher than 40 ppb in the metropolitan areas of Seoul, Busan, Gwangju, and Daegu (in Figures 3 and 4). In August, it was higher than 40 ppb only in Seoul. These cities are high-density metropolitan areas with a high population density, few green areas, a high road density,

and a high proportion of urbanized areas. In contrast, areas with low $NO_2$ concentrations, regardless of the season, were mainly in high altitudes and faced low developmental pressure.

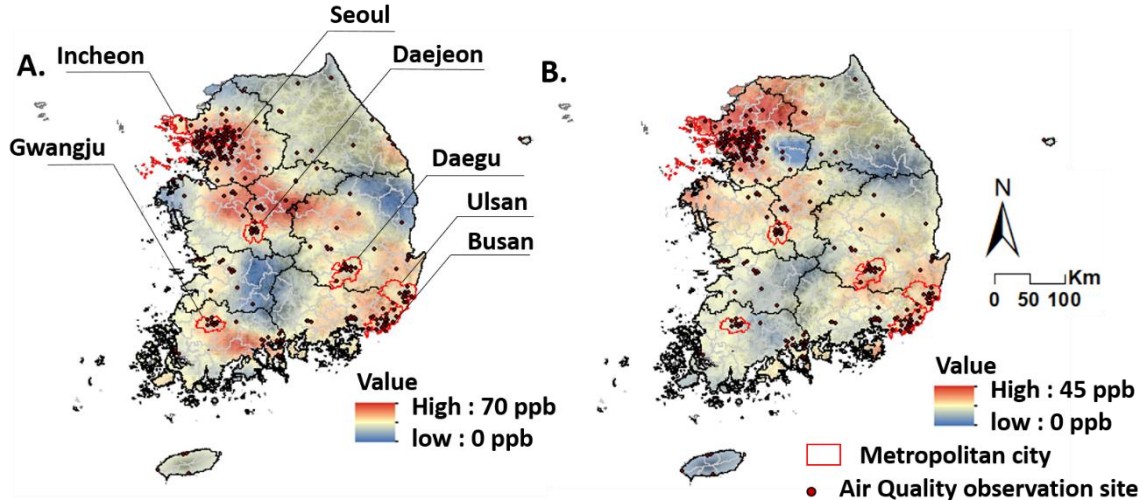

**Figure 3.** Concentration map of $NO_2$, produced by cokriging. a: Winter (Feb.). b: Summer (Aug.).

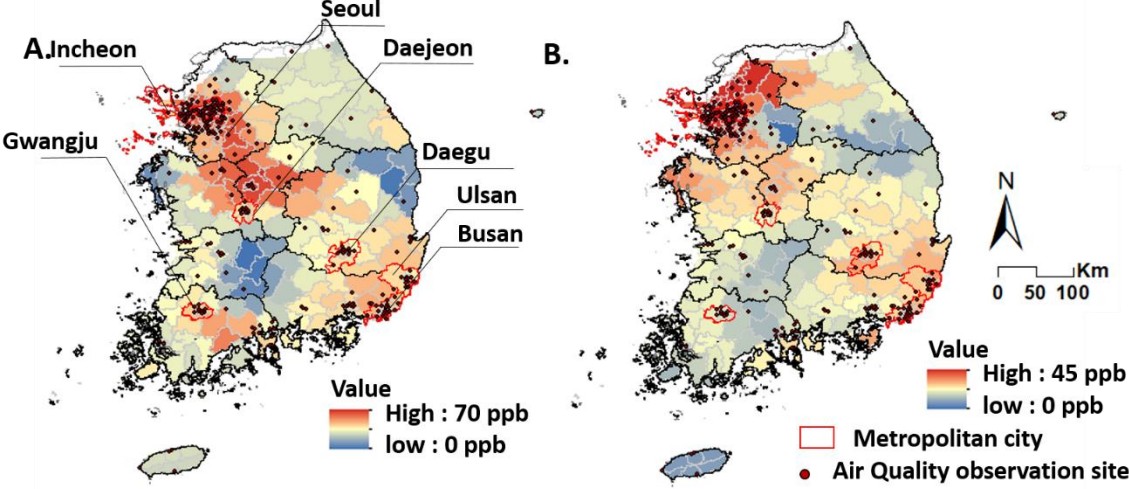

**Figure 4.** $NO_2$ concentration map by administrative district. a: Winter (Feb.). b: Summer (Aug.).

To produce the $NO_2$ cell unit data that was used to construct the $NO_2$ map for administrative units, spatial interpolation was conducted using satellite images based on observation information. Before implementing the cokriging process, a semivariogram was obtained using the $NO_2$ concentration data taken from 281 ground stations and satellite data taken via GOME-2 and DEM. Since the data were not biased, spatial interpolation was performed without any data conversion. The lag size for the February data was 0.258, while that for the August data was 0.241; both had a nugget of (0,0). Because the accuracy of the interpolation methods did not differ significantly, the most common method, ordinary cokriging, was used.

According to the cross-validation results with the observed and interpolated data, the slope of the regression line for the winter $NO_2$ concentrations was 0.661 and that for summer was 0.574 (Figure 5). The RMSS values were 0.917 (in Figure 5, left side) in winter and 0.775 in summer (in Figure 5, right side) (The closer the RMSS error is to 1, the higher the accuracy of spatial interpolation). As a result of cokriging, the coefficient of the regression line was 0.5 or more for both seasons (summer: 0.574, winter: 0.661; See Figure A2 for semivariograms).

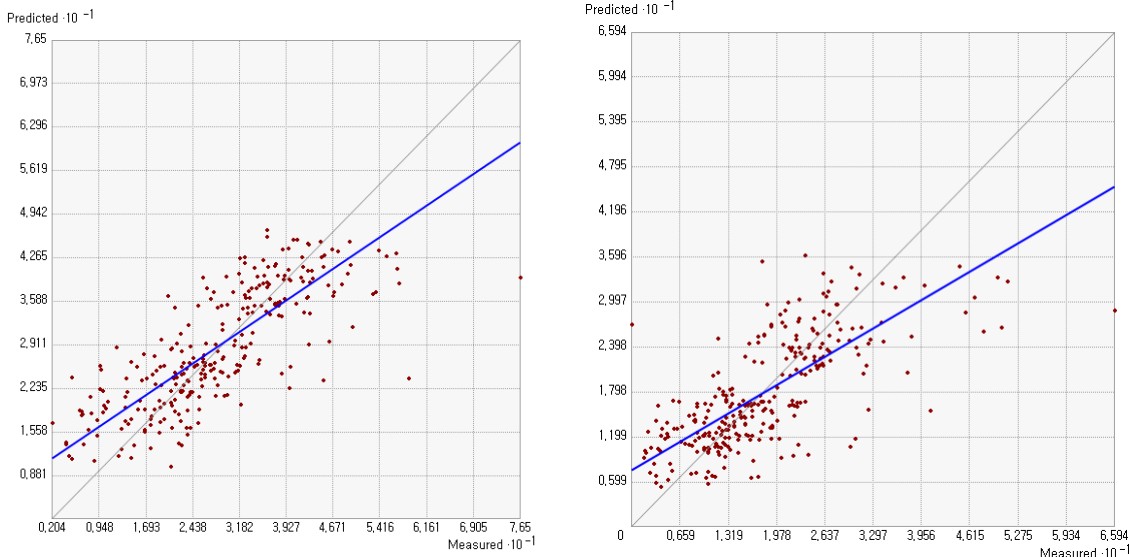

**Figure 5.** Scatter plot of the observed versus predicted $NO_2$ concentrations, constructed using a cross-validation comparison. Left: Winter (Feb.). Right: Summer (Aug.).

### 3.2. Analysis of the Accuracy of Nationwide $NO_2$ Concentrations Measured by Administrative Unit

To compare the difference between the observed and interpolated values using the cokriging method, their statistical differences were compared. $NO_2$ concentration values were found to be 4.82–76.75 ppb in winter (average 29.48 ± 11.62 ppb) and 2.01–76.05 ppb in summer (average 20.65 ± 10.96 ppb). The satellite image values were 0–65.71 ppb in winter (average 36.44 ± 17.91 ppb) and 0–63.60 ppb in summer (average 32.79 ± 17.11 ppb). The results of spatial interpolation with cokriging were 0–57.32 ppb $NO_2$ in winter (average 33.50 ± 11.12 ppb) and 0–48.69 ppb $NO_2$ in summer (average 25.14 ± 10.65 ppb). When the spatial interpolation results were converted into those for administrative units, the $NO_2$ concentration was 0–52.10 ppb in winter (average 32.52 ± 11.41 ppb) and 0–48.30 ppb in summer (average 24.52 ± 10.80 ppb).

To assess the accuracy of the $NO_2$ concentration maps constructed by administrative unit, we statistically compared the maps produced using the following data sources: (a) observations, (b) satellite images, (c) interpolated data via cokriging, and (d) cokriging data averaged by administrative unit. It was found that the map produced using cokriging data averaged by administrative unit reduced the uncertainty in the observation data and the satellite images and increased the accuracy. For all four data sources, the winter $NO_2$ concentrations were higher than the summer levels because the use of fossil fuels increases in winter due to the need for heating. The highest standard deviation (17.91 ppb) was observed for the remote sensing data for winter and the lowest value (10.65 ppb) for the interpolated data via cokriging for summer. The standard deviation of satellite image data was larger than those of other data. This is because, since the satellite images were included in the cell type data for all areas, there was a large difference in values compared with the observation data which were crowded within the city. Compared with the average value of the observed data, the satellite image value was higher by about 23.64% in winter and 58.82% in summer. However, the data generated by the administrative unit was higher by about 10.32% in winter and about 18.75% in summer than the observation data (Figure 6). The reason the average of the spatially interpolated values is larger than the average of the observed values is the satellite image values used in the interpolation are larger than the observed values.

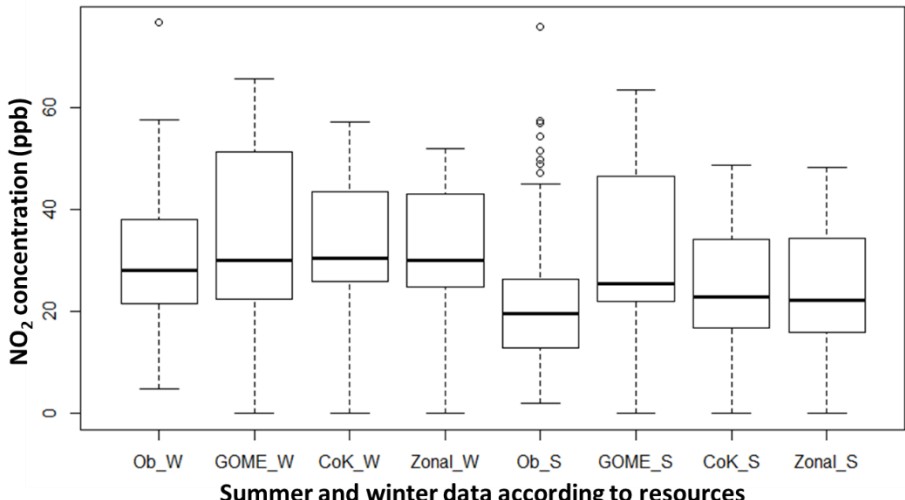

**Figure 6.** Comparison of NO$_2$ concentration data by season and data source (Ob: observation, GOME: GOME-2 remote sensing, CoK: Interpolation using the cokriging method, Zonal: averaged CoK value by administrative unit, w: winter, s: summer).

### 3.3. Analysis of NO$_2$ Concentration Values Using LUR Models

To determine which environmental variables should be employed in the LUR models, correlation analysis was conducted for the 18 environmental variables and NO$_2$ concentration. Environmental variables that showed a high correlation with NO$_2$ concentration and low correlation with other variables were selected. Of these, 11 variables had a correlation coefficient of ±0.5 or stronger: NDVI, Road_Length, R_Forest, R_Urban, R_Res, R_Com, R_Cul, R_Pub, R_Coni, R_Pop, and RL/Pop. NDVI was significantly correlated with R_Forest, R_Field, R_Paddy, R_Coni, and R_Mixed at the 1% level, while the correlation coefficients between RL_Pop and Road_Length, R_Urban, R_Res, R_Com, R_Cul, R_Pub, R_Coni, and R_Pop were 0.7 or higher. Therefore, NDVI and RL_Pop, which had little or no correlation with each other or with the season dummy variable, were selected for use in the LUR model as an NO$_2$ reduction factor and emission source, respectively.

The performance of the LUR models are summarized in Table 2. The final LUR models consisted of two predictors: the NDVI as an NO$_2$ reduction factor and road length per unit population (RL_Pop) as an NO$_2$ source. In a previous study that developed an LUR model for NO$_2$, the R$^2$ ranged from 0.44 to 0.96 [37]. In this study, the adjusted R$^2$ values of the LUR models were 0.335 and 0.526 for urban and non-urban regions, respectively.

The NDVI was negatively correlated with NO$_2$ concentration in the LUR models for both urban and non-urban regions. The effect of the NDVI per unit area on the reduction in NO$_2$ levels was about 2.34% higher in non-urban areas than in urban areas. The average NDVI values for urban and non-urban areas were 0.39 (± 0.15) and 0.56 (± 0.20), respectively. The reduction in NO$_2$ levels due to NDVI was similar for urban and non-urban regions based on the standardized beta coefficients of the LUR models (urban regions: −0.460; non-urban regions: −0.471). However, NO$_2$ concentrations in urban areas were much higher than those in non-urban areas. Therefore, the unstandardized beta coefficients for NDVI were −29.385 in the urban and −22.593 in the non-urban areas. Therefore, if the vitality of urban vegetation increases, then the reduction effect of the NDVI in urban areas will be higher.

The road length per unit population (RL_Pop) was positively correlated with NO$_2$ levels in the LUR models. The standardized coefficients for RL_Pop in non-urban regions were 43.51% larger than those in urban regions. The average road length per unit population was about 8.81 (± 2.41) km/person in urban regions and about 2.34 (± 2.01) km/person in non-urban regions.

**Table 2.** Land use regression model for urban and non-urban regions.

| Urban Regions | | | | | | (Adj. $R^2$ = 0.335) | |
|---|---|---|---|---|---|---|---|
| Model | Unstandardized Coefficients | | Standardized Coefficients | $t$ | Sig. | Collinearity Statistics | |
| | B | Std. Error | Beta | | | Tolerance | VIF |
| (Constant) | 41.360 | 3.700 | | 11.179 | 0.000 | | |
| NDVI | −29.385 | 5.409 | −0.460 | −5.433 | 0.000 | 0.918 | 1.089 |
| RL_Pop | 3.565 | 1.155 | 0.261 | 3.087 | 0.003 | 0.918 | 1.089 |
| Non-Urban Regions | | | | | | (Adj. $R^2$ = 0.526) | |
| Model | Unstandardized Coefficients | | Standardized Coefficients | $t$ | Sig. | Collinearity Statistics | |
| | B | Std. Error | Beta | | | Tolerance | VIF |
| (Constant) | 28.403 | 1.184 | | 23.998 | 0.000 | | |
| NDVI | −22.593 | 1.689 | −0.471 | −13.378 | 0.000 | 0.954 | 1.048 |
| RL_Pop | 9.185 | 0.699 | 0.462 | 13.131 | 0.000 | 0.954 | 1.048 |

## 4. Discussion

To generate a highly accurate $NO_2$ concentration map for administrative units within South Korea, this study employed the cokriging method, which combines point-type observation data and satellite images to produce $NO_2$ concentration data for individual cells at the national level. In this study, the spatial interpolation results using cokriging were accurate because information was not lacking for any area owing to the combined use of point-based spatial data and cell-unit-based satellite image data. This method has been used for the spatial interpolation of branch information for large areas in various fields [16,34]. The ordinary spatial interpolation method that uses only observation point information tends to overestimate the observation point information because the weights of the observation points are simply derived based on distance [51]. In addition, since the observation points are too densely concentrated in the city, information on non-urban area becomes lacking; thus, when spatial interpolation is performed, the variation in the $NO_2$ concentration distribution may become distorted. Therefore, spatial interpolation performed using both satellite image data and observation data yields more accurate results. However, cokriging has not previously been employed for spatial interpolation that utilizes observation data and satellite images to generate $NO_2$ concentration data. We confirmed the feasibility of using this method in this study, and we believe that it will be useful for generating precise spatial data for national or global units.

When we analyzed $NO_2$ levels by season and region to determine the accuracy and applicability of our proposed approach, the accuracy of the data was found to be high even if the data was processed by administrative unit. The means and standard deviations of the four data sources (observations, satellite imagery, spatial interpolation using both observations and satellite imagery, and spatial interpolation within administrative units) were compared to determine their level of agreement on $NO_2$ concentrations. The average $NO_2$ concentration by administrative unit was the closest to the observed values, and the average concentration calculated using spatial interpolation based on the geostatistical method was slightly larger than that for the administrative units. The observations and satellite imagery produced the lowest and highest standard deviation, respectively, while the standard deviations of the spatially interpolated data and the data averaged by administrative unit were similar to each other. In this study, we used corrected images provided by NASA but on comparing the measured $NO_2$ concentrations with the satellite images, the difference between the standard deviation and the mean was found to be larger than that for the other data sources. In other words, the $NO_2$ concentrations calculated based on the geostatistical spatial interpolation method used in this study

were closer to the observations than when only satellite images were used. It is thus possible to overcome the limitations of observation data without using homogeneous survey point information. Previous research has only compared the accuracy of satellite images and observations [30,52,53], and large variations in $R^2$ values depending on the type of satellite image and spatial scale have been reported [53]. Therefore, we expect that our proposed method for measuring $NO_2$ concentration using geostatistical spatial interpolation will provide more accurate information than approaches based on a combination of observation data and satellite imagery, in regions with insufficient observation points.

To reduce $NO_2$ concentrations at the national level, it is necessary to have access to accurate information on $NO_2$ concentrations in defined administrative units because decisions are made within these administrative units (e.g., in prioritizing areas and allocating budgets). The $NO_2$ concentration map by administrative unit constructed in this study, using $NO_2$ concentrations in cell units derived via the cokriging method, is potentially useful in this regard because it can overcome the disadvantages of observation data and satellite images individually. In this study, the differences in $NO_2$ concentration between the observation data, satellite image data, and spatial interpolation were statistically compared. Spatial interpolation by administrative unit produced $NO_2$ concentrations that were more consistent with the observation data, with fewer outliers than seen in the observation data alone. In addition, although the observation data only reports $NO_2$ concentrations at the observation point, the $NO_2$ concentrations for individual administrative units after spatial interpolation were proven to have high accuracy because locations that did not have a measurement station nearby were analyzed using satellite images.

In this study, LUR models were developed using the $NO_2$ concentrations for administrative units including urban and non-urban areas nationwide. Urban and non-urban areas vary greatly in terms of land cover, including the proportion of man-made areas, due to the different intensities of human activity. Therefore, the generation and reduction of $NO_2$ differ for these areas, and the $R^2$ value for these areas differs. The $R^2$ values of LUR models in previous studies have been reported to vary greatly depending on the region, but the accuracy for urban regions was always higher than that for non-urban regions. Bechle et al. [54] developed an LUR model using environmental variables for both urban and rural areas. The adjusted $R^2$ value for urban areas was 0.76 and that for rural areas was 0.49. The LUR model was more accurate in urban and suburban regions than in rural regions, which means that the differences in $NO_2$ concentration caused by environmental variables were not significant in natural areas. In Reference [55], when the prediction of $NO_2$ concentration was classified by region, the accuracy was low due to the small number of measuring stations in the suburbs compared with urban areas. However, in this study, the urban regions were mostly developed areas with a very high density. Because the $NO_2$ concentrations and predictor variables were all generated within administrative units, $NO_2$ concentrations across an individual administrative unit were high due to the influence of roads in the urban area. Therefore, the $R^2$ values of the LUR model for the urban and non-urban regions were 0.335 and 0.526, respectively. The non-urban areas had a higher $R^2$ value because of the variety of land use rates, and the relationship between the $NO_2$ concentration and the environmental variables was clear. In the future, if a city is analyzed using a smaller spatial unit than administrative districts (*gu*), it is expected that the $R^2$ value will increase further.

In this study, LUR models were developed considering both the factors that reduce $NO_2$ concentration, such as NDVI, and cause $NO_2$ concentration increases, such as RL_Pop, in an administrative unit. RL_Pop represents the road length per unit population and this variable is used as a predictor variable to represent emission sources because it has a very strong correlation with other environmental variables. NDVI indicates the degree of vitality of the vegetation, and in the LUR model developed in this study, the vegetation with high vitality has the effect of reducing $NO_2$ concentration. In previous studies that have developed LUR models based on $NO_2$ concentration, all road-related variables were used and there were also differences in the spatial units analyzed. In addition, previous LUR models have been developed mostly for cities by collecting data for houses, blocks, and cells in accordance with the size of the city. Different environmental variables in the same

city could be classified using data such as road length [1,55], road type [19–22], cell-specific population, and distance from residence. Because these previous studies were based on observational information, in many cases, the relationship between $NO_2$ concentration and the environmental variables was analyzed by generating circular buffers of various radii for the environmental variables [39,56]. The environmental variables used in previous research and in this study thus differed due to the different spatial scales at which the trends were analyzed. It is therefore necessary to consider data on both a city and national scale when developing LUR models for the purpose of reducing $NO_2$ levels in the future.

## 5. Conclusions

Accurate $NO_2$ concentration maps for administrative units are required for informed policy decisions regarding air pollution control. In this study, $NO_2$ concentration maps for February and August 2010 that are based on administrative units were generated using the cokriging method for on 240 observation points and satellite images. Additionally, we have developed LUR models for urban and non-urban areas using the $NO_2$ concentration data generated. With this model, the influences of environmental variables (RL_Pop, NDVI) on $NO_2$ concentration was confirmed. If we manage urban $NO_2$ concentrations using the two environmental variables derived from the LUR model ("road density per population (RL_Pop)" as an air pollutant source, and "NDVI" as an air pollutant absorption medium), we can increase the sustainability of the cities.

We developed the LUR model using various environmental variables related to land use, but we did not consider weather-related variables in this study. In this study, the monthly average $NO_2$ concentration was used; since meteorological variables affect the $NO_2$ concentration in units of seconds, they were not used because the time scale was not appropriate. However, meteorological phenomena such as rainfall may have a significant effect on $NO_2$ concentration, so it is necessary to reflect these environmental variables in the development of future LUR models. In addition, we did not consider the intermediate production process because we created a detailed $NO_2$ concentration map for the present $NO_2$ concentration levels. However, since $NO_2$ can be generated by secondary reactions in addition to in automobile exhaust gas, it should be studied in the future.

Despite this limitation, this study was able to prove the feasibility of producing a highly accurate $NO_2$ concentration map by administrative unit using satellite images and observation data. It also demonstrates that $NO_2$ concentrations could be predicted by developing an LUR model based on this map. This approach can be useful in predicting $NO_2$ concentrations for decision-making related to environmental policy in countries where $NO_2$ concentration data is lacking.

**Author Contributions:** J.R. planned the entire study, analyzed the data, and wrote the paper. C.P. and S.W.J. discussed the structure and content of the thesis. All authors substantially revised the first draft after discussing the structure and the results of this paper.

**Funding:** This research was funded by Korea Environment Industry & Technology Institute(KEITI) through "The Chemical Accident Prevention Technology Development Project" project, funded by Korea Ministry of Environment (MOE), grant number "No. 2016001970001", the National Research Foundation of Korea (NRF) through "A Study on Optimization and Model Development for the Assessment of Ecosystem Services for Fine Dust Reduction" Project, grant number "NRF-2018R1D1A1B07049160", and by a Korea University.

**Conflicts of Interest:** The authors declare no conflicts of interest.

## Appendix A

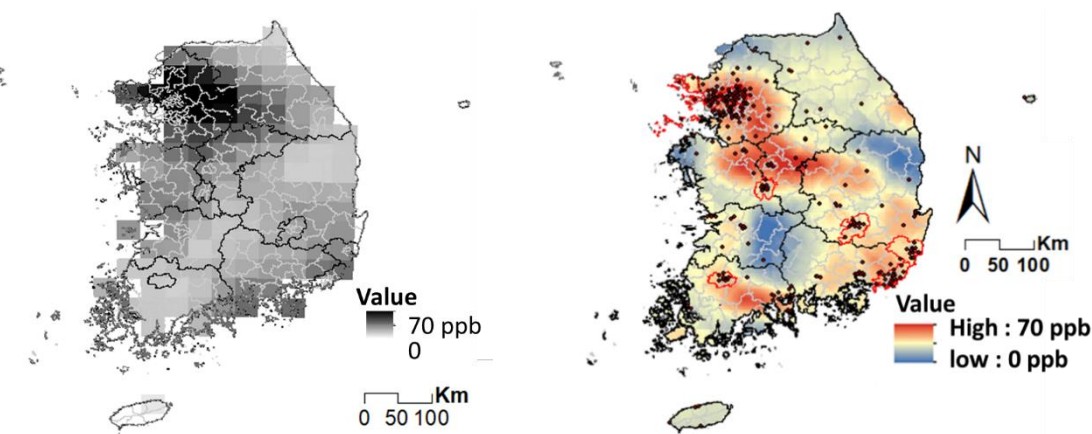

**Figure A1.** Comparison of spatial resolution between remote sensing data and spatial interpolation data (Left: GOME-2 remote sensing data in winter (Feb.); right: Spatial interpolation with cokriging in winter (Feb.)).

## Appendix B

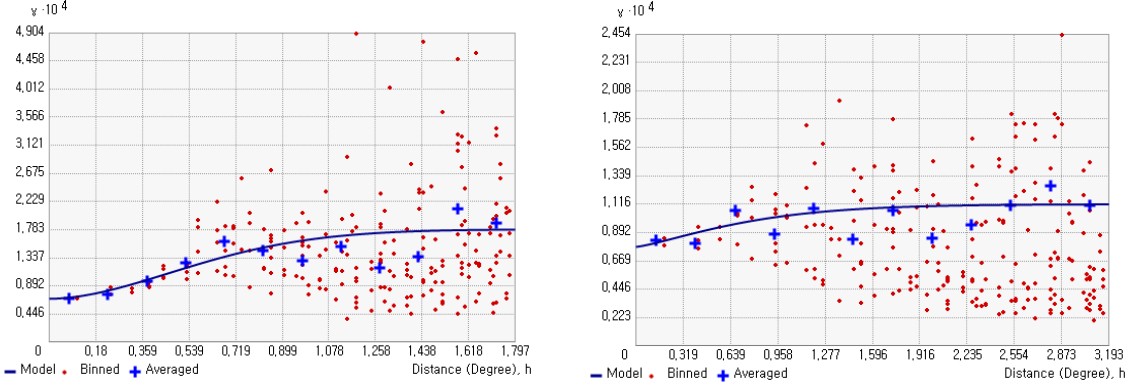

**Figure A2.** Semivariogram (Left: winter (Feb.); right: summer (Aug.)).

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
