# Peer review of "Mapping and Statistical Analysis of NO2 Concentration for Local Government Air Quality Regulation"

_sustainability, doi:10.3390/su11143809_

Round 1
Reviewer 1 Report
Dear Authors,
The paper “Mapping and statistical analysis of NO2 3 concentration by administrative unit” deals with a very interesting topic and adapt for the Journal of Sustainability. In order to publish please consider the following minor requests:
1) Introduction: it is well written but it would be better if there were a state of the art about LUR models. There are several works about this methodology. Please refer to:
- Famoso et al. (2017). Measurement and modeling of ground-level ozone concentration in Catania, Italy using biophysical remote sensing and GIS. International Journal of Applied Engineering Research, 12(21), 10551-10562.
- Hoek et al. (2008). A review of land-use regression models to assess spatial variation of outdoor air pollution. Atmospheric Environment, 42(33), 7561-7578.
- Cowie et al. (2019). Comparison of model estimates from an intra-city land use regression model with a national satellite-LUR and a regional Bayesian Maximum Entropy model, in estimating NO 2 for a birth cohort in Sydney, Australia. Environmental Research, 174, 24-34.
2) Figure one: despite the fact its well presented, i would appreciate if it was larger to better understand the spatial distribution locations.
3) Figure two: This graph is confusing. It would be better to present air pollutants with the same unit of measure.
Best Regards
Author Response
We would like to express our appreciation for your detailed and helpful comments on our manuscript “Mapping and statistical analysis of NO2 concentration by administrative unit”. Your advice has been very helpful in helping me to describe the differentiation of my thesis and to revise the limitations and findings so that I can clearly communicate them to the reader. Based on your comments, we have revised the manuscript carefully. We have attempted to incorporate your comments into the article as accurately as possible. If you note any further shortcomings, we welcome your comments. We hope the revised manuscript will meet the standard of publication of Sustainability.
Please check an attached file.

Reviewer 2 Report
The study by Ryu et al. reports the use of cokriging technique for interpolating concentrations of NO2 in the Republic of South Korea.
The study seems to be properly conducted; however, as a statistical analysis, the authors are invited to attach the raw data, or to add additional supplementary material in order to make it repeatable.
The conclusions are weak: it was found a (relatively?) high correlation between the NO2 concentrations, interpolated by cokriging and averaged by administrative unit with, and the observed values. For these reasons, the study lacks significance, and I don't believe that it is appropriate for being published in Sustainability.
Moreover,
The title is ambiguos. In particular, the part "by administrative unit" it is not meaningful, because neither in the title, abstract, keywords, and Introduction, has been reported that it refers specifically to "administrative units" in Korea.
25 - Please define LUR
Fig 1 - The insert in Fig 1 should be separated from the picture, as they represent two different things and it could be misleading.
Part of Paragraph 2.1 belongs to the Introduction section.
212 - The text reports "previous studies," but only one study has been cited.
Table 1 - Why "Percentage of industrial area" and "Percentage of urbanization area" are not log-transformed?
246 - How the P<0.01 was chosen?
247 - Why 0.5 was used as a threshold? For which indicators correlation coefficient was greater than 0.5? In that case which variable would be dropped?
282 - "High" compared to? It would be appreciated that R2 would be reported as well.
299 - The affirmation is ambiguous.
In the introduction, the direct effect of NO2 is reported; however, it has not been reported that it reacts with other chemicals in the air to form secondary pollutants, which is of a main concern as well.
Please, fix all the subscripts (i.e. NO2).
Can you please report variograms?
Part of the conclusions is redundant and they should be shortened.
All the references are not corresponding. Please fix them.
Author Response
We would like to express our appreciation for your detailed and helpful comments on our manuscript “Mapping and statistical analysis of NO2 concentration by administrative unit”. Your advice has been very helpful in helping me to describe the differentiation of my thesis and to revise the limitations and findings so that I can clearly communicate them to the reader. Based on your comments, we have revised the manuscript carefully. We have attempted to incorporate your comments into the article as accurately as possible. If you note any further shortcomings, we welcome your comments. We hope the revised manuscript will meet the standard of publication of Sustainability.
Please see the attachment.

Reviewer 3 Report
Referee comment on “Mapping and statistical analysis of NO2 concentration by administrative unit” by Jieun Ryu et al. submitted to Sustainability
Ryu et al. constructed a detailed NO2 concentration map for the administrative units in South Korea. The used data sources and methodology are solid, and the results are clearly delivered. It is generally well written, and the topic is within the scope of Sustainability. I recommend the publication of this study after the following comments can be addressed.
Major comments:
(1) Line 25: based on your results (Fig.7 and Line 362-364), it is not correct that “the average concentrations were highest for the administrative units” in Abstract.
(2) Figure 1: the data used to generate this global NO2 distribution should be explained in the main text.
(3) Line 140-141: I suggest adding figures to show the NO2 distributions based on the satellite data you chosen as the satellite map is compared with other three NO2 distribution maps.
(4) Table 1: In the column of “Other comments”, why in most cases “Log” is used while in a few cases “Log” is not used?
(5) Line 291: why the average standard error was higher for August comparing with that in February?
(6) Line 304-305: why the data averaged by administrative unit are higher than the observed values? Is this because of the bias of the satellite data?
(7) Line 350-352: I suggest discussing what is the difference between the interpolation results for areas with dense surface observation points and the areas with few surface observations but satellite datesets largely used?
Minor comments:
(1) Line 25: Add “Land Use Regression” before “LUR”.
(2) Line 34: Reference [1] is missing in the reference lists.
(3) I could not search out references [2], [9], [10], and [11] from the published papers. Are they in Korean published in local journals?
(4) Several references did not match the content in the main text, e.g., [24] on Line 120, [25] on Line 128, [32] on Line 185, and [39] on Line 228.
(5) Equation (1): Z should not be capitalized.
(6) Equation (2): What is the difference between Z and Zi?
(7) Line 180: delete “a” in front of LUR models. Also for Line 392.
(8) Equation (4): Should x1 be xn? Are α, β, and γ different for different environmental variables? Then α, β, and γ should be αn, βn, and γn.
(9) Line 213: add references after “previous studies” which also applied the 18 variables.
(10) Line 224: Antanasijevic et al., 2018 should be [38].
(11) Line 232: References should be added after the DEM model to explain it.
(12) Line 312: Section 3.2 here should be Section 3.3.
(13) Line 316-318: I can not see this result from Table 2.
(14) Line 421: should be “environmental” variables.
(15) Line 422: “past studies” should be “previous studies”.
(16) Line 445: “RMSE” should be “RMSS”.
Author Response
We would like to express our appreciation for your detailed and helpful comments on our manuscript “Mapping and statistical analysis of NO2 concentration by administrative unit. Your advice has been very helpful in helping me to describe the differentiation of my thesis and to revise the limitations and findings so that I can clearly communicate them to the reader. Based on your comments, we have revised the manuscript carefully. We have attempted to incorporate your comments into the article as accurately as possible. If you note any further shortcomings, we welcome your comments. We hope the revised manuscript will meet the standard of publication of Sustainability.
Please see the attachment.

Round 2
Reviewer 2 Report
The author satisfactorily answered to all the concerns addressed.